# USTAM: UNIFIED SPATIO-TEMPORAL ATTENTION MIXFORMER FOR VISUAL OBJECT TRACKING

## ABSTRACT

In this paper, we present a unified spatio-temporal attention MixFormer framework for visual object tracking. Within the vision transformer framework, we design a cohesive network consisting of target template and search region feature extraction, cross-attention utilizing spatial and temporal information, and task-specific heads, all operating in an end-to-end manner. Incorporating spatial and temporal attention modules within the network enables simultaneous feature extraction and emphasis, allowing the model to concentrate on target-specific discriminative features despite changes in illumination, occlusion, scale, camera pose, and background clutter. Stacking multiple non-hierarchical blocks allows meaningful features to be extracted while irrelevant features are discarded from the provided target template and search region. The simultaneous spatio-temporal attention module is employed to accentuate target appearance features and alleviate variation in the object state across frame sequences. Qualitative and quantitative analysis, including ablation tests based on various tracking benchmarks, validates the robustness of the proposed tracking methodology.

## 1 INTRODUCTION

Visual object tracking (VOT) is a traditional computer vision application that tracks the position of a target object in space or across different camera viewpoints. VOT has been successfully employed in various applications, but its performance can still be hindered by changes in illumination, scale, camera pose, occlusion, and background clutter. In this context, VOT methods that employ deep neural networks tend to outperform conventional approaches. In particular, VOT using vision transformers (ViTs) (Dosovitskiy et al., 2021) is no longer merely considered an alternative to convolutional neural networks (CNNs) for VOT but rather has become the leading candidate for superior performance in visual object tasks. ViT based VOT involves three main steps: (1) utilizing a ViT-based backbone to extract features from the target template and search region, (2) employing a cross-attention module to integrate features between the target template and search region, and (3) employing task-specific heads to precisely localize the target and estimate its bounding box. There has been a trend in VOT to use a ViT-based backbone to extract features and emphasize them by adding annexed encoders and decoders via the embedding of numerous attention modules. However, these methods lead to greater model complexity and incur higher computational costs. As a result, it is essential to find the appropriate balance between effective feature extraction and the incorporation of attention modules. ViT based VOT models also lack explicit modeling of the relationship between spatial, which is related to the appearance of the object and assists in target localization, and temporal information, which records changes in the state of the object between frames.

In this paper, we introduce a novel approach to VOT networks by designing a unified integration of visual feature extraction and spatio-temporal cross-attention information within a single module. The spatio-temporal attention mechanism is trained to identify target features from a given search region. The temporal attention generated from the previous search sequence for the same target object is employed in the current search region, followed by spatial attention that reinforces any significant features. The spatial attention module highlights appearance information within the present search region, while the temporal block distills insights from previous time steps within the search region. Ultimately, predictions are generated by weighting and combining current and past information to produce a comprehensive understanding of the scene. Recognizing that the positional information of the target in previous frame $t-1$ can be advantageous in determining its position in current frame

$t$, our proposed tracker employs the attention map from the previous frame to guide the target position in the present frame. Furthermore, we propose the unified spatio-temporal attention MixFormer (USTAM) model, which amplifies significant targets from the search features to bolster tracking performance. As demonstrated using experimental validation, this approach places fewer demands on computational resources and can be readily applied to various attention modules.

The main contributions of the proposed method can be summarized as follows:

- We present a simple but effective unified VOT pipeline for feature extraction, target information integration, and localization estimation within the framework of a ViT network.
- The spatio-temporal attention module in the network is designed to emphasize target appearance localization in the change of the environment.
- We propose an end-to-end VOT network to effectively extract discriminative features from the MixFormer pipeline. The experiment results quantitatively and qualitatively verify the robustness of the proposed approach.

## 2 RELATED WORKS

**CNN-based trackers.** Most CNN-based VOT methods aim to provide an improved target representation via hierarchical feature representation learning on larger-scale datasets (Pu et al., 2018; Gundogdu & Alatan, 2018), instead of utilizing pre-training (Han et al., 2018; Wu et al., 2019), formulating numerous target models to capture a variety of target characteristics (Li et al., 2018c; Wang et al., 2018), enhancing the generalizability of models with spatial-temporal information (Li et al., 2018b; 2020), or fusing different deep features to complement semantic and spatial information (Bhat et al., 2018; Ma et al., 2019). Zhang et al. Zhang et al. (2020) proposed an object-aware anchor-free network (Ocean) consisting of an offline anchor-free component and an online model update component. In another study, Xie et al. (2022) presented a target-dependent feature network (SBT-B) that learns cross-image feature correlation through multiple layers. Siamese networks (SNNs) (Bromley et al., 1993), which use two identical CNN branches, have been employed for VOT because SNNs offer improved performance with a relatively low computational complexity. Many SNN-based VOT approaches have appeared (Bertinetto et al., 2016; Li et al., 2018a; 2019; Chen et al., 2020; Guo et al., 2021; Cheng et al., 2021; Voigtlaender et al., 2020). For example, Bertinetto et al. (2016) introduced a fully-convolutional Siamese VOT network (SiamFC) trained in an end-to-end manner, while Li et al. (2018a) proposed a Siamese region proposal network (SiamRPN) consisting of an SNN for feature extraction and a region proposal network for regression and the classification of targets. SiamRPN was extended to SiamRPN++ (Li et al., 2019) with training on ResNet and both layer-wise and depth-wise aggregation to improve its accuracy. In another study, Chen et al. (2020) introduced a Siamese box adaptive network incorporating a fully convolutional network into an SNN (TransT), while Voigtlaender et al. (2020) presented a re-detection strategy using two-stage object detection network. Guo et al. (2021) attempted to solve the problems associated with a pre-fixed target region size and global matching by proposing target-aware region selection that adopts a graph attention mechanism. Cheng et al. (2021) also designed an SNN with two modules: a relation detector for filtering distractors and a refinement module for more precise tracking. Similarly, Mayer et al. (2021) proposed a target candidate association network (KeepTrack) robust to distractor. SNN-based approaches aim to reduce the computational costs of CNN-based VOT methods for real-time applications.

**CNN-transformer-based trackers.** A number of researchers have conbined CNNs with transformers, with most of these hybrid CNN-transformer-based trackers adopting an SNN as the feature-extracting backbone and feeding the extracted features into the transformer to obtain similar features for the target in the search region. Wang et al. (2021) was the first to propose a CNN-transformer VOT approach, which they referred to as TrDiMP, in which features from both template patches and the search region are extracted using a CNN, with the template features fed into the encoder and the search region features fed into the decoder of the transformer. Similarly, Yu et al. (2021) presented a CNN-transformer tracker using an encoder-decoder transformer, and Yang et al. (2023) reduced the computational cost of TrDiMP by adopting a deformable transformer. Chen et al. (2021) suggested a CNN-transformer network consisting of three modules: a CNN as a backbone, a transformer for feature fusing, and a prediction network. Zhong et al. (2022) employed feature fusion by incorporating a correlation module into a transformer network. Zhao et al. (2021) also used a transformer

to capture global information from target templates, and the learned global features were utilized as cues to calculate the correlation between the target and search region. Yan et al. (2021) proposed VOT based on the object detection transformer STARK, which captures both the spatial and temporal cues of the target. Mayer et al. (2022) proposed VOT with a transformer-based model prediction module (ToMP) in which features from the target templates and search regions were extracted from the CNN and concatenated, with both features jointly fed into the encoder and decoder of the transformer. Gao et al. (2022) designed an attention-in-attention (AiATrack) module by incorporating an inner attention module into a transformer network to focus on appropriate correlations and ignore erroneous ones.

**Fully transformer-based trackers.** Because CNN-transformer trackers use a CNN as the backbone for feature extraction, it is difficult for these hybrid CNN-transformer VOT methods to capture global features, thus research has started to focus on global information learned by transformers. Fully transformer-based trackers can be classified into two-stream transformer trackers, in which one transformer acts as the backbone and the other is used to discover relationships, and one-stream transformer trackers, which employ a single transformer. Xie et al. (2021) proposed a VOT method using dual transformers for the target template and search region. The two transformers consisted of local and global attention blocks, and the output features of the dual branches were fed to a cross-attention block to calculate the relationship between the target and the search region. Lin et al. (2022) adopted Swin Transformer (Liu et al., 2021), which was originally designed as a general-purpose backbone, to propose SwinTrack as a means to improve the interactions in learning the features of the target template and search region. SwinTrack has three modules: a feature extraction transformer, a feature fusion transformer, and a prediction module. Cui et al. (2022) presented MixFormer as a transformer tracking method whose core component is a mixed attention module (MAM) used to extract features and integrate target information simultaneously. Ye et al. (2022) also proposed a one-stream tracker that unifies feature extraction and feature fusion using a transformer with an early candidate elimination module to improve the inference efficiency (OSTrack). A problem associated with one-stream trackers is that they often unnecessarily calculate the attention between template patches and all search patches, and Gao et al. (2023) attempted to overcome this by introducing a generalized relation model based on the adaptive selection of search region patches. Xie et al. (2023) also attempted to fully utilize both temporal and spatial information within a one-stream transformer tracker. Wu et al. (2023) employed two cores: a masked autoencoder in the tracking network to capture both spatial and correlated spatial information and an attention dropout mechanism to restrict the within-frame token interactions. Wei et al. (2023) proposed ARTrack, an encoder-decoder transformer without a prediction head, which solves VOT as a coordinate sequence interpretation problem.

## 3 USTAM: UNIFIED SPATIO-TEMPORAL ATTENTION MIXFORMER

In this section, we present the USTAM model, a ViT-based VOT method within an updated search area by integrating temporal and spatial information from target-specific features into a coupled VOT framework as shown in Fig. 1. Motivated by MixFormer (Cui et al., 2022), which unifies

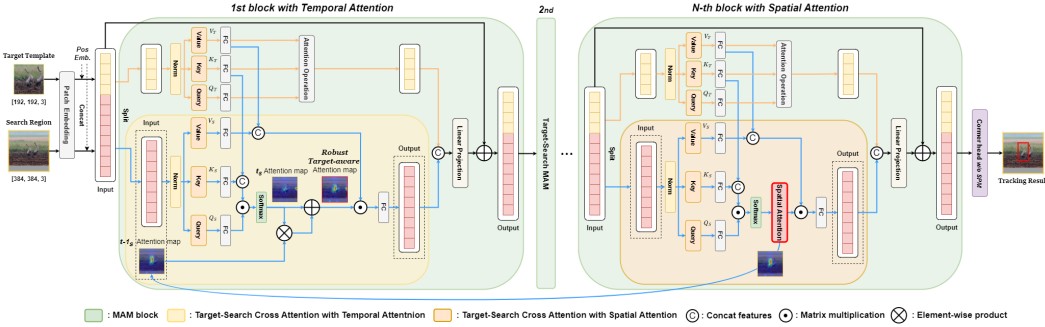

Figure 1: Proposed USTAM model for VOT involving the embedding of a mixed attention module and a bounding box prediction head.

feature extraction and target information integration using an MAM for target-specific feature extraction and extensive communication between the target and search area, we design an end-to-end tracking model. This model primarily consists of an iterative MAM-based backbone with an embedded spatio-temporal feature emphasis module and a bounding box prediction head. To effectively and efficiently unify target-based search area detection and spatio-temporal information communication, we adopt an asymmetric attention scheme for each MAM block. In the first MAM block of the backbone, we devise a temporal information update mechanism, while we incorporate spatial attention into the last MAM block. We refer to the combination of spatial and temporal attention as *Spatio-Temporal attention*. This is expected to facilitate the effective extraction of target and background information. The spatial attention map of the last block is the updated temporal information and serves as input for the first block in the subsequent step. The remaining MAM blocks maintain a consistent structure.

### 3.1 TEMPORAL ATTENTION

Given target template $I_T$ and search area $I_S^t$ at time $t \geq 0$, the proposed network detects target bounding box $B^t$, with search area $I_S^{t+1}$ automatically adjusted by $B^t$ and fed as input into the following $t + 1$ time step. The network consists of iterative MAM blocks as its backbone, and temporal attention is installed in the first block. The information on which attention is focused for the target at time $t - 1$ is guided at time $t$ so that attention is focused on the target from among similar objects and confusing background. The MAM block is used to unify feature extraction and information integration by adopting the target (or self-) attention and search (or cross-) attention. With the query, key, and value $(Q_{tg}, K_{tg}, V_{tg})$ and $(Q_s, K_s, V_s)$ at time $t$ for target and search attention, respectively, the mixed key and value are generated via concatenation as follows:

$$K_m = Concat(K_{tg}, K_s), \quad V_m = Concat(V_{tg}, V_s), \tag{1}$$

and the asymmetric attention feature map at time $t$ is defined as:

$$Attention_{tg}^t = softmax\Big(\frac{Q_{tg}K_{tg}^T}{\sqrt{d}}\Big)V_{tg}, \quad Attention_s^t = softmax\Big(\frac{Q_s K_m^T}{\sqrt{d}}\Big)V_m, \tag{2}$$

where $d$ is the dimension of the key. Because the target in temporal motion moves continuously without sudden jumps, meaning that the location of the target at the current time is close to its location at the immediately preceding time step, we apply temporal attention to the search attention of the first block. For notational simplicity, $G^t$ is used to represent the attention map at time $t$ as:

$$G^t := softmax\Big(\frac{Q_s K_m^T}{\sqrt{d}}\Big). \tag{3}$$

Temporal attention is then defined to modify the search attention feature map given in Eq. 2 as:

$$Attention_{s-t}^t = \Big[G^t \otimes G_{sp}^{t-1} + G^t\Big]V_m = \Big[G^t \otimes (G_{sp}^{t-1} + 1)\Big]V_m, \tag{4}$$

where $G_{sp}^{t-1}$ is the spatial attention map in the last MAM block at time $t - 1$, and the notation $\otimes$ denotes the element-wise product (or Hadamard) of the matrices. As explained in the following subsection, the spatial attention map $G_{sp}^{t-1}$ can distinguish the target from the background so that $G^t \otimes G_{sp}^{t-1}$ in Eq. 4 has the effect of strengthening the location information for the target, while the target-specific features are preserved by again adding $G^t$ to it, $G^t \otimes G_{sp}^{t-1} + G^t$ in Eq. 4.

### 3.2 SPATIAL ATTENTION

We employ spatial attention in the last MAM block to distinguish the target from the complex background, while the network integrates the target information. The search attention in the final block generates map $G_f^t$ at time $t$,

$$G_f^t = softmax\Big(\frac{Q_{s,f}K_{m,f}^T}{\sqrt{d}}\Big), \tag{5}$$

for search query $Q_{s,f}$ and mixed key $K_{m,f}$ obtained from the block. This attention map reflects the similarity between the query row vectors and key row vectors calculated using the inner product. That is, the attention map has higher similarity values in target-related rows in a complex background and low similarity values in background-background or background-target related rows. In order to enhance these properties further, the spatial attention is placed in the search attention before applying mixed value $V_{m,f}$. The spatial attention takes $G_f^t = (g_{ij})_{i,j=1}^n$ (with $n$ the number of tokens related to the search area) as input and produces map $G_{sp}^t$ of the relative average score given by:

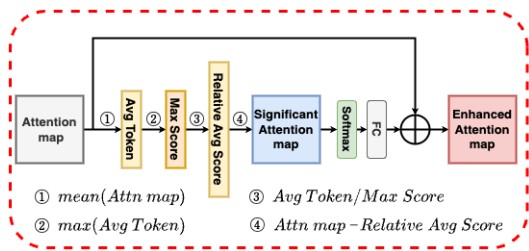

Figure 2: Spatial attention used to emphasize the meaningful features and remove the noisy features from a given image.

$$\left(G_{sp}^t\right)_{i,j} = g_{i,j} - m_{ave}\left(\frac{1}{n}\sum_{i=1}^n g_{ij}\right), \qquad (6)$$

where $m_{ave}$ represents the largest of the mean values in each row of $G_f^t$, which is given in parentheses on the right side of Eq. 6. The spatial attention then sequentially applies softmax, FC, and residual connection to map $G_{sp}^t$ to finally obtain an enhanced attention map. This process using the relatively average score given in Eq. 6 experimentally results in the strengthening of the target-related features and the weakening of the background-related features. The structure of the spatial attention module is presented in Fig. 2.

## 3.3 TRAINING AND INFERENCE

We predict the target bounding box given the top left and bottom right vertices of the box. The bounding box is the output of the bounding box prediction head fed by the feature map from the final MAM block. The search area $I_S^{t+1}$ is given as twice the width and height of the bounding box centered on the box, and it is used as input for the next step at time $t + 1$ along with target template $I_T$. We train the proposed tracking network according to the standard training process for conventional trackers (Chen et al., 2021; Cui et al., 2022; Yan et al., 2021), and we optimize the model using loss function $\mathcal{L}$, given as:

$$\mathcal{L} = \mathcal{L}_1(U, V) + \lambda \mathcal{L}_{GIoU}((U, V)), \qquad (7)$$

for ground-truth bounding box $U$ and predicted bounding box $V$ for the target. Here, $\mathcal{L}_1$ is the $L_1$-loss and $\mathcal{L}_{GIoU}$ is the GIoU loss (Rezatofighi et al., 2019) defined as:

$$\mathcal{L}_{GIoU}((U, V)) = 1 - GIoU(U, V) = 1 - \left(\frac{|U \cap V|}{|U \cup V|} - \frac{|C \setminus (U \cap V)|}{|C|}\right) \qquad (8)$$

for the smallest box $C$ containing $U \cap V$. Parameter $\lambda$ is the trade-off weight.

## 4 EXPERIMENTAL RESULTS

This section presents the results of experiments comparing the proposed approach with current state-of-the-art (SOTA) VOT methods using public datasets. We also visualize the attention maps of the target objects to easily understand how the proposed network and attention modules work. Ablation analysis is also employed to verify the robustness of the proposed approach by analyzing the effects of the spatial and temporal attention modules in the backbone network.

**Experiment setup.** The proposed method is implemented on a PC with Intel(R) Core$^{TM}$ i7-7700 CPU (3.66 GHz) and an NVIDIA(R) A100 GPU in Python using the PyTorch framework. To ensure a fair performance evaluation for VOT with SOTA methods, we analyze their VOT performance using the TrackingNet (Müller et al., 2018), LaSOT (Fan et al., 2019), and GOT-10k (Huang et al., 2021) datasets. We measured the accuracy (AUC), normalized precision ($P_{Norm}$), and precision (P) for the LaSOT and TrackingNet datasets, and the average overlap (AO), $SR_{50}$, $SR_{75}$ for the GOT-10k

Table 1: The hyperparameters used in the experiments. All of the parameters are equally applied to all models for a fair evaluation of their performance.

| Learning rate | 1e-4 (initial) , 1e-5 (epoch 400) |
|---|---|
| Optimizer | AdamW with weight decay $10^{-4}$ |
| Search region image size | L : $384 \times 384$, B : $288 \times 288$ |
| Target template image size | L : $192 \times 192$, B : $128 \times 128$ |
| Trade-off parameter | 0.4 |

Table 2: Performance comparison of state-of-the-art VOT models using the LaSOT, TrackingNet and GOT-10k datasets. Where * denotes for tracker only trained on GOT-10k. The two best-performing methods are highlighted in red and blue, respectively.

| Tracker | Published | LaSOT | | | TrackingNet | | | GOT-10k | | |
|---|---|---|---|---|---|---|---|---|---|---|
| | | AUC (%) | $P_{Norm}$(%) | P (%) | AUC (%) | $P_{Norm}$(%) | P (%) | AO (%) | $SR_{50}$(%) | $SR_{75}$(%) |
| Ocean | ECCV20 | 56.0 | 65.1 | 56.6 | - | - | - | 61.1 | 72.1 | 47.3 |
| TransT | CVPR21 | 64.9 | 73.8 | 69.0 | 81.4 | 86.7 | 80.3 | 67.1 | 76.8 | 60.9 |
| STARK | ICCV21 | 67.1 | 77.0 | - | 82.0 | 86.9 | - | 68.8 | 78.1 | 64.1 |
| KeepTrack | ICCV21 | 67.1 | 77.2 | 70.2 | - | - | - | - | - | - |
| SwinTrack-B | NeurIPS22 | 69.6 | 78.6 | 74.1 | 82.5 | 87.0 | 80.4 | 69.4 | 78.0 | 64.3 |
| SBT-B | CVPR22 | 65.9 | - | 70.0 | - | - | - | 69.9 | 80.4 | 63.6 |
| AiATrack | ECCV22 | 69.0 | 79.4 | 73.8 | 82.7 | 87.8 | 80.4 | 69.6 | 80.0 | 63.2 |
| ToMP | CVPR22 | 67.6 | 78.0 | 72.2 | 81.2 | 86.2 | 78.6 | - | - | - |
| MixFormer-L | CVPR22 | 70.1 | 79.9 | 76.3 | 83.9 | 88.9 | 83.1 | 75.6 | 85.7 | 72.8 |
| OSTrack-B$_{384}$ | ECCV22 | 71.1 | 81.1 | 77.6 | 83.9 | 88.5 | 83.2 | 73.7* | 83.2* | 70.8* |
| ARTrack-B$_{384}$ | CVPR23 | 72.6 | 81.7 | 79.1 | 85.1 | 89.1 | 84.8 | 75.5* | 84.3* | 74.3* |
| USTAM-L | Ours | 72.0 | 81.1 | 78.8 | 84.7 | 89.0 | 84.8 | 75.2 | 83.8 | 74.4 |

dataset as the metrics for quantitative evaluation and comparison. The AO measures the accuracy of a tracking algorithm by calculating the average intersection-over-union (IoU) overlap between the predicted bounding boxes and the ground truth bounding boxes across all frames in the dataset. $SR_{50}$ and $SR_{75}$ represent a tracking algorithm's ability to successfully track target objects with IoU overlap thresholds of 0.50 and 0.75 or higher, respectively. Table 1 presents the hyperparameters used for the training process. To ensure a fair evaluation and comparison, all parameters were equally set to large and base models[1].

## 4.1 QUANTITATIVE ANALYSIS

Table 2 presents a comparison of the results for the LaSOT, TrackingNet, and GOT-10k datasets obtained from our proposed method and the SOTA VOT methods. The performance of the proposed method is competitive across all datasets, which is attributed to the emphasis on spatial and temporal features extracted from the search region image, which accounts for changes in illumination, scale, occlusion, camera pose, and other variables. Our method produces a good performance for all three datasets with little overfitting because the backbones of both USTAM-Base and USTAM-Large are initialized with the parameters for the Masked Autoencoder (He et al., 2022) pretrained on the large ImageNet dataset. Table 2 also illustrates that there is an overall improvement in performance resulting from the transition from CNN-based algorithms to ViT-based models. This improvement is achieved by considering global and local features from both the target template and the search region.

**LaSOT dataset.** The LaSOT dataset is one of the largest and most densely annotated tracking benchmark datasets. ARTrack, which is based on a plain ViT encoder, achieves the best performance by employing an additional decoder that uses previous estimates (i.e., spatio-temporal prompts) as command tokens. Although the AUC values for the proposed method are 0.6% lower than those of ARTrack, our method achieves the second-best results by simultaneously emphasizing spatial and temporal features even when the target object disappears in a video sequence. Thus, our method

---
[1]Source code will be released on our GitHub account

effectively mitigates irrelevant noise by preserving spatio-temporal features. In contrast, existing methods often have difficulty locating meaningful patches because they tend to lose significant features for the target and may erroneously emphasize incorrect features or accumulate errors following the target's reappearance in the scene.

**TrackingNet dataset.** TrackingNet is a comprehensive dataset and a benchmark for object tracking in natural settings. The proposed method produces competitive results for this dataset, achieving the second-best results (about 0.5% lower than ARTrack) despite only employing an encoder using spatio-temporal attention without the assistance of a decoder. The results from TrackingNet, as presented in Table 2, demonstrate the superiority of ViT-based algorithms over their CNN-based counterparts. This is attributed to the enhanced ability of ViTs to detect both global and local features. Our method, which utilizes a ViT backbone network, successfully preserves the global and local features of the target object while also emphasizing its crucial attributes. The proposed model thus proves its usefulness for transformer-based models with an attention module because it reliably tracks objects under various environments.

**GOT-10k dataset.** The GOT-10k dataset contains over 10,000 video segments featuring real-world moving objects, along with more than 1.5 million manually labeled bounding boxes. While Mix-Former achieves the best performance in terms of the AO and $SR_{50}$ metrics by utilizing the flexibility of an MAM for simultaneous feature extraction and target information integration. Our USTAM-L, MixFormer-L, OSTrack$_{384}$, and ARTrack$_{384}$ exhibit an improved AO performance by over 5% compared to previous algorithms due to the use of a ViT in the VOT.

## 4.2 QUALITATIVE ANALYSIS

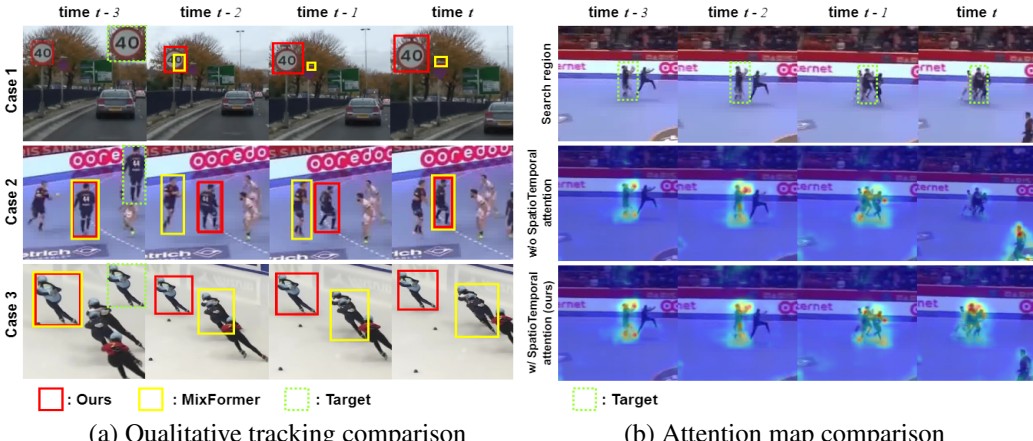

(a) Qualitative tracking comparison

(b) Attention map comparison

Figure 3: Example of the sequential tracking results (left) from our proposed approach (red box) and MixFormer (yellow box). Target is indicated by the dotted green box at time $t - 3$. Attention maps (right) in the last block of backbone w/ and w/o spatio-temporal attention from GOT-10k. Target is indicated by the dotted green box in the search region.

Figure 3(a) presents a comparison between the tracking results for the proposed USTAM model and MixFormer, an SOTA approach to VOT. In the first case in Fig. 3(a), rapid changes in scale within the scene can result in tracking failure, even when the target template is distinguishable from the background. However, the proposed method employs temporal attention to consistently transmit the object's information to the model over time, leading to an enhancement in object tracking performance. In the second and third cases, objects with similar appearances undergo partial occlusion, resulting in tracking failure due to the presence of similar features. However, in the proposed method, it becomes apparent that emphasizing the object's features enables stable tracking even in scenarios where objects share similar colors and shapes. To understand the underlying reasons for the effective performance of the proposed algorithm across various environments, attention maps are employed for visualization in Fig. 3(b). The absence of a spatio-temporal attention module leads to a tendency to emphasize the wrong objects with similar characteristics, leading to tracking failure. However, the

spatio-temporal attention module enhances the tracking performance by simultaneously considering location and appearance information even in the presence of diverse environmental changes.

## 4.3 ABLATION STUDY

To verify the robustness of the proposed USTAM tracker, we conduct the ablation study. Table 3 presents the evaluation results for the models trained with the GOT-10k dataset. While the AO for USTAM-Base is 2.9% lower than that of USTAM-Large, the evaluation and comparison results demonstrate that the proposed method is less dependent on the training data. Furthermore, Table 3 reveals that the tracking algorithms continue to be influenced by the size of the search region. Using the same USTAM-Base model, we only increase the search region image size from 288 to 384, and the AO performance improves by 2.3%. ARTrack, which utilizes an autoregressive model with a general encoder-decoder architecture, achieves the best evaluation performance by eliminating the customized heads and post-processing to simplify the tracking pipeline.

Table 3: Performance comparison for state-of-the-art models using the GOT-10k dataset. Tracker* is trained on the GOT-10k training set only. The two best-performing methods are highlighted in red and blue respectively.

| Trackers | Search region image size | GOT-10k | | |
| --- | --- | --- | --- | --- |
| | | AO | SR$_{50}$ | SR$_{75}$ |
| STARK-ST101* | 320 | 68.8 | 78.1 | 64.1 |
| SwinTrack-B* | 384 | 69.4 | 78.0 | 64.3 |
| MixFormer-22k* | 320 | 70.7 | 80.0 | 67.8 |
| MixFormer-1k* | 320 | 71.2 | 79.9 | 65.8 |
| TATrack-B* | 224 | 73.0 | 83.3 | 68.5 |
| ARTrack-B* | 256 | 73.5 | 82.2 | 70.9 |
| USTAM-B* | 288 | 70.8 | 80.8 | 65.9 |
| USTAM-B* | 384 | 73.1 | 82.5 | 69.5 |

Table 4 presents an analysis of how the spatial and temporal attention modules within the network impact a range of performance factors. When spatial attention is added to the network, there is a slight difference in terms of speed and parameters. In both the base and large models, the MACs metric increases slightly by 0.4 G and 3G, respectively. However, the performance of the base model trained with only the GOT-10k dataset improves by 1.2%, and the LaSOT dataset with the large model trained on the entire dataset improves by 1.7%. By also adding temporal attention, the GOT-10k and LaSOT base models improve by 0.7% and 1.6% respectively, despite minimal changes in speed, MACs, and the parameters. When both spatial attention and temporal attention are applied, the performance with the GOT-10k dataset improves by 2.3% and by 1.9% on the LaSOT dataset with a minimal difference in speed, MACs, and parameters. As a result, applying both spatial attention and temporal attention simultaneously leads to the highest model performance. We conduct

Table 4: Comparison of the speed, MACs, and parameter performance according to spatial and temporal attention using the GOT-10k and LaSOT datasets

| Trackers | Spatial attention | Temporal attention | Speed(A100) (FPS) | MACs (G) | Params (M) | AO (%) |
| --- | --- | --- | --- | --- | --- | --- |
| USTAM-Base*$_{288}$ | ✓ | ✓ | 47.1 | 43.6 | 97.2 | **70.8** |
| | ✓ | | 47.2 | 43.6 | 97.2 | 69.7 |
| | | ✓ | 47.3 | 43.2 | 97.1 | 69.2 |
| | | | 47.4 | 43.2 | 97.1 | 68.5 |
| Trackers | Spatial attention | Temporal attention | Speed(A100) (FPS) | MACs (G) | Params (M) | AUC (%) |
| USTAM-Large | ✓ | ✓ | 30.1 | 274.6 | 318.0 | **72.0** |
| | ✓ | | 30.4 | 274.6 | 318.0 | 71.8 |
| | | ✓ | 30.3 | 271.6 | 317.7 | 71.7 |
| | | | 31.1 | 271.6 | 317.7 | 70.1 |

Table 5: Ablation study of the effectiveness by embedding spatio-temporal attention in the the previous VOT models.

| Model | Spatio-temporal attention | GOT-10k | | |
|---|---|---|---|---|
| | | AO | SR$_{50}$ | SR$_{75}$ |
| SwinTrack-Tiny* | | 66.5 | 75.9 | 60.2 |
| | ✓ | **68.1** | **78.0** | **61.8** |
| MixFormer-Base*$_{288}$ | | 68.5 | 77.7 | 64.3 |
| | ✓ | **70.8** | **80.8** | **65.9** |
| MixFormer-Base*$_{384}$ | | 72.3 | 81.5 | 69.3 |
| | ✓ | **73.1** | **82.5** | **69.5** |

experiments with and without the spatio-temporal attention on the GOT-10k dataset to demonstrate the performance and flexible applicability of the spatio-temporal attention module. The spatio-temporal attention module is attached next to the backbone in the encoder-decoder module of the SwinTrack-Tiny and MixFormer-Base models. As shown in Table 5, with the spatio-temporal attention, the SwinTrack-Tiny model performance improves by 1.6% and the MixFormer-Base models improve by 2.3% and 0.8%, respectively. This verifies the flexibility of the spatio-temporal attention module for use in any model to enhance performance.

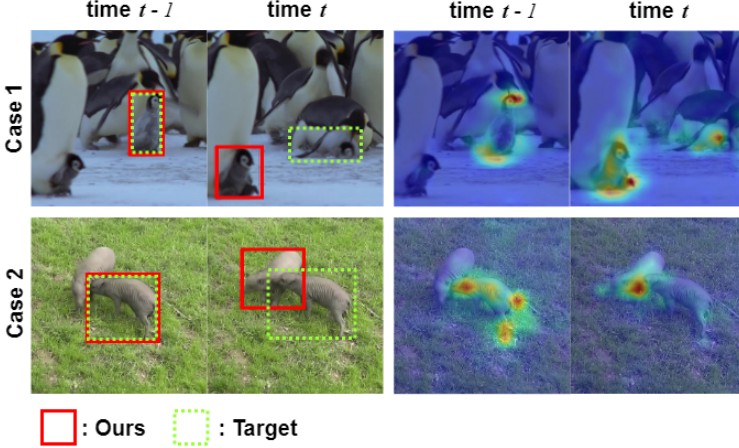

Figure 4: Visualization of incorrect sequential tracking results (left) and attention maps (right) from the GOT-10k dataset. The target and our results are indicated by the dotted green box and red box in the images, respectively.

## 5 DISCUSSION

In this paper, we present a straightforward yet effective unified VOT network. Our proposed USTAM is designed as an end-to-end VOT network to effectively extract discriminative features from the MixFormer pipeline. The modularized spatial and temporal attention simultaneously emphasize the target's appearance and localization, efficiently eliminating redundant and irrelevant features within the search region. The quantitative and qualitative analysis, including ablation test, demonstrate that the proposed USTAM model can be effectively employed in various real-world applications. However, the proposed USTAM model still has notable limitations, such as incorrect feature extraction, partial occlusion, and similar object overlapping. Figure 4 presents examples of qualitative error related to the predicted target, which may be caused by the influence of environmental changes. In further research, we plan to work on emphasizing meaningful patches and eliminating noisy patches through lightweight modeling and knowledge distillation techniques in VOT.

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
