# OpenReview forum: "USTAM: UNIFIED SPATIO-TEMPORAL ATTENTION MIXFORMER FOR VISUAL OBJECT TRACKING"
_ICLR.cc/2024/Conference — Submitted to ICLR 2024_

### Official Review · Reviewer_yHrZ · 2023-10-30

**Soundness:** 2 fair
**Presentation:** 2 fair
**Contribution:** 2 fair
**Rating:** 5
**Confidence:** 4

**Summary:**

The paper introduces a spatial-temporal attention MixFormer framework for visual object tracking, with experimental results affirming its effectiveness.

**Strengths:**

The USTAM approach is crafted as an end-to-end VOT network, integrating spatial and temporal attentions.

**Weaknesses:**

The proposed approach is an incremental improvement of MixFormer tracker. Many of the components in the paper such as MAM block,
asymmetric attention, loss function have already been proposed in MixFormer.

**Questions:**

1. It would be beneficial to allocate a dedicated section to MAM, considering it serves as the primary building block for this approach. Distinguishing between the author's specific contributions and those stemming from MAM can be challenging otherwise.
2. Rearranging the reference section in order of the last name of the first author would enhance searchability.
3. The dimensions of G_f^i nxn don't seem to align. G_f^i represents the attention map between the search area and mixed, which is the combination of both the search and target areas.
4. Figure 1 appears too small to discern the letters effectively.
5. In equation (6), given that g_i,j is the attention map in equation (5), it follows that the sum of each row of G_f_t should be 1 after applying softmax. This operation essentially subtracts a constant value. Could you elaborate on the rationale behind this step?

---

> ### Author Response · Authors · 2023-11-20
>
> Answer of question (1) Thank you for your kind comment. Contrary to what you understood, all blocks in our network are composed of MAM blocks. However, we apply temporal attention using the attention map of the search region from the previous frame, and we introduce spatial attention in the last MAM block to enhance attention for the target. This way, we leverage spatiotemporal information to improve tracking performance.
>
> Answer of question (2) Thank you for your kind suggestion. We have crafted the reference section following the guidelines provided by ICLR to enhance searchability. I hope it proves helpful in your searches.
>
> Answer of question (3) Thank you for your valuable comment. To explain further, during the attention operation where the Query comes from the search region and the Key and Value come from the mixed (i.e., mixed targets and search region), the resulting attention map is utilized to apply spatial attention. In this case, we specifically use the attention values corresponding to the search region, and thus, the dimension is nxn.
>
> Answer of question (4) Thank you for your comment. In Figure 1, we have done our best to effectively portray the overall architecture of the proposed network. While it may not be efficient for character identification, we have prioritized the visual representation as the image is more crucial than the characters. We would appreciate it if you could focus more on the visual aspects rather than the characters.
>
> Answer of question (5) We are very sorry to confuse you with the typo in equation (6). We changed  j=1 to i=1. Softmax was applied to rows and the average was obtained for columns. The score that each token has on all tokens was calculated by averaging the columns.

---

### Official Review · Reviewer_ciuV · 2023-10-31

**Soundness:** 2 fair
**Presentation:** 2 fair
**Contribution:** 1 poor
**Rating:** 3
**Confidence:** 5

**Summary:**

This paper proposes to track the target object using spatial and temporal attention-based Transformer networks. This paper points out that existing works fail to find the appropriate balance between effective feature extraction and the incorporation of attention modules. They also lack explicit modeling of the relationship between spatial and temporal information. The experiments are conducted based on three widely SOT datasets.

the issues of this work are that:

the idea of incorporating attention mechanisms into the Transformer networks for tracking is not new;
the speed of this tracker is about 30-40+ FPS, which is not fast compared with other SOTA trackers, such as OSTrack;
Considering the limited novelties and regular tracking efficiency, I tend to reject this paper.

**Strengths:**

This paper proposes to track the target object using spatial and temporal attention-based Transformer networks. This paper points out that existing works fail to find the appropriate balance between effective feature extraction and the incorporation of attention modules. They also lack explicit modeling of the relationship between spatial and temporal information. The experiments are conducted based on three widely SOT datasets.

**Weaknesses:**

the issues of this work are that:

the idea of incorporating attention mechanisms into the Transformer networks for tracking is not new;
the speed of this tracker is about 30-40+ FPS, which is not fast compared with other SOTA trackers, such as OSTrack;
Considering the limited novelties and regular tracking efficiency, I tend to reject this paper.

**Questions:**

1. re-organization of the novelties proposed in this work;
2. showing the real advantages of this work;

---

> ### Author Response · Authors · 2023-11-20
>
> Answer of question (1) Let me reiterate the novelties we propose in this paper. In the MixFormer, which we use as a baseline, additional training is employed with a score prediction module for online target update, utilizing temporal information. However, instead of using this module, we enhance performance by applying temporal attention that only a very slight increase in speed and parameters. To enhance the reliability of temporal attention, we apply spatial attention to the last block of the transformer for the previous frame. In this manner, we propose spatial and temporal attention that leverages spatiotemporal information to enhance tracking performance.
>
> Answer of question (2) Let me explain you the real advantages of our work. As demonstrated in the qualitative analysis in Section 4.2 of the paper, even when the scale of the target changes in a frame or when there are similar objects around the target, our proposed spatial attention strengthens the attention map for the target. Subsequently, through temporal attention in the next frame, we guide the model with positional information from the previous frame, resulting in a more robust tracking performance.

---

> > ### Comment · Reviewer_ciuV · 2023-11-23
> >
> > Thanks for your response. After reviewing the comments from other reviewers I still think the current work is not ready for publication on iclr.

---

### Official Review · Reviewer_DDtH · 2023-11-09

**Soundness:** 2 fair
**Presentation:** 2 fair
**Contribution:** 2 fair
**Rating:** 5
**Confidence:** 5

**Summary:**

This paper proposes a unified spatio-temporal attention mixformer framework for video object tracking (VOT). Specifically, they’re two main contributions stated by the authors: 1) a simple yet effective unified pipeline is proposed for feature extraction, target information integration, and localization estimation within the framework of a ViT network; 2) a spatio-temporal attention module is introduced to more effectively distinguish the target from the complicated background. Experimental results on several popular VOT benchmarks show the proposed approach performs favorably against SOTA trackers.

**Strengths:**

- The idea seems to be somewhat effective, which can observe some performance improvements on the main VOT benchmarks (e.g., LaSOT and TrackingNet).
- The paper is well organized, which is easy to follow.
- Sufficient related works are discussed in Sec. 2.

**Weaknesses:**

- The statement for ‘We present a simple but effective unified VOT pipeline for feature extraction, target information integration, and localization estimation within the framework of a ViT network’ is not really true. This unified framework has already been proposed in previous one-stage trackers, e.g., OSTrack, including all the feature extraction, target interaction and localization in the same ViT framework.
- The contribution in this paper is somewhat incremental. It seems that the proposed framework is still similar to the MixFormer framework, although it uses a ViT-based architecture and considering the previous target state by using the temporal attention module.
- The usage of the temporal attention module is a bit similar to use the Cosine Window (e.g., also used in OSTrack), which also makes the tracker object moves smoothly in consecutive frames. In this paper, the authors make it in a learnable way by using the attention map in the previous frame. But one unsolved problem is about the reliability of the previous target state. If the previous prediction is noisy, the effectiveness of the proposed approach is also questionable.
- Missing some essential details and unfair comparison. It is not clear whether the proposed tracker use the pre-trained models. e.g., OSTrack and DropTrack. In Table 2, the authors compare with the OSTrack-384 only trained on GOT-10k training split, while the proposed approach additionally  uses more training data, which is not fair. From Table 3, it seems that the proposed USTAM-B-384 trained on GOT-10K is inferior to OSTrack-384. What’s the reason? Does the compared two approaches use the same pre-trained model?

**Questions:**

- The statement for ‘We present a simple but effective unified VOT pipeline for feature extraction, target information integration, and localization estimation within the framework of a ViT network’ is not really true. This unified framework has already been proposed in previous one-stage trackers, e.g., OSTrack, including all the feature extraction, target interaction and localization in the same ViT framework.
- The contribution in this paper is somewhat incremental. It seems that the proposed framework is still similar to the MixFormer framework, although it uses a ViT-based architecture and considering the previous target state by using the temporal attention module.
- The usage of the temporal attention module is a bit similar to use the Cosine Window (e.g., also used in OSTrack), which also makes the tracker object moves smoothly in consecutive frames. In this paper, the authors make it in a learnable way by using the attention map in the previous frame. But one unsolved problem is about the reliability of the previous target state. If the previous prediction is noisy, the effectiveness of the proposed approach is also questionable.
- Missing some essential details and unfair comparison. It is not clear whether the proposed tracker use the pre-trained models. e.g., OSTrack and DropTrack. In Table 2, the authors compare with the OSTrack-384 only trained on GOT-10k training split, while the proposed approach additionally  uses more training data, which is not fair. From Table 3, it seems that the proposed USTAM-B-384 trained on GOT-10K is inferior to OSTrack-384. What’s the reason? Does the compared two approaches use the same pre-trained model?

---

> ### Author Response · Authors · 2023-11-20
>
> Answer of question (1) Thank you for your good point. As you said, the unified ViT framework has indeed enhanced performance in the realm of visual object tracking. However, methods for effective binding and treatment of spatio-temporal features have not been studied in the unified ViT framework. This paper distinguishes itself from existing SOTA works by not only presenting a novel method but also substantiating its efficacy through experimental validation.
>
> Answer of question (2) Thank you for your comment. There is a problem that the networks are similar because we use MixFormer as a baseline. However, the existing MixFormer uses a score prediction module for online target update to utilize temporal information. However, this requires additional learning. However, we propose a temporal attention module that simply but effectively utilizes temporal information without the need for additional learning.
>
> Answer of question (3) Thank you for your good comment. Since the prediction of the previous target may be noisy, there is a problem that the reliability of the prediction of the previous target state may decrease. To solve this, we apply a spatial attention module to the last layer of the transformer for the previous frame and a temporal attention module to the first layer of the transformer for the current frame to increase the reliability of predictions about the target state.
>
> Answer of question (4) Thank you for your comment. We prepared a performance table in Table-2 as compared in other papers. However, there was no indication of the GOT-10k model in Table-2, so it was updated. The GOT-10K model is marked with *. The performance of our GOT-10k model was compared in table-3 of the ablation study section.

---

### Author Response · Authors · 2023-11-20

I want to express appreciation to all the reviewers for their valuable feedback and suggestions. Your insights have been extremely helpful in revising and enhancing the paper. I am truly grateful for the time and effort you have devoted to reviewing the manuscript. Your comments have played a vital role in shaping the final version of the paper. Once again, thank you sincerely for your invaluable contributions.

---

### Meta-Review · Area_Chair_UjQi · 2023-12-06

**Metareview:**

The paper was considered by 3 reviewers.  The major concerns were:

1. Incremental novelty - a unified framework has already been proposed in OSTrack. A similar framework to MixFormer is used. [DDtH]
2. Incremental novelty - the temporal attention module is similar to the cosine window (used by OSTrack) [DDtH]
3. How to handle unreliable target states in the temporal attention module? [DDtH]
4. Missing details in the experiments, perhaps unfair comparison. [DDtH]
5. Why is USTAM-384 worse than OSTrack on GOT-10k? [DDtH]
6. limited novelty, limited efficiency [ciuV]
7. What is the advantage of the work? [ciuV]
8. Presentation issues about MAM and authors contribution. [yHrZ]
9. Presentation issues about the equations. [yHrZ]

The authors wrote a response. However, the author response did not assuage the concerns of the reviewers, in particular about the incremental novelty and the performance comparison. The AC agrees with the concerns and thus recommends reject.

**Justification For Why Not Higher Score:**

Incremental novelty, and performance worse than OSTrack when compared fairly w/ the same training set.

**Justification For Why Not Lower Score:**

n/a

---

### Decision · Program_Chairs · 2024-01-16

Reject